# Elucidation of the Molecular Mechanism of Wet Granulation for Pharmaceutical Standard Formulations in a High-Speed Shear Mixer Using Near-Infrared Spectroscopy

**DOI:** 10.3390/ph13090226

**Published:** 2020-08-31

**Authors:** Ryo Omata, Yusuke Hattori, Tetsuo Sasaki, Tomoaki Sakamoto, Makoto Otsuka

**Affiliations:** 1Faculty of Pharmacy, Musashino University, 1-1-20 Shinmachi, Nishi-Tokyo, Tokyo 202-8585, Japan; s1143045@stu.musashino-u.ac.jp (R.O.); yhattori@musashino-u.ac.jp (Y.H.); 2Graduate School of Medical Photonics, Shizuoka University, 3-5-1 Jyohoku, Naka-ku, Hamamatsu 432-8561, Shizuoka, Japan; sasaki.tetsuo@shizuoka.ac.jp; 3Division of Drugs, National Institute of Health Sciences, 3-25-26 Tonomachi, Kawasaki-ku, Kawasaki City 210-9501, Kanagawa, Japan; tsakamot@nihs.go.jp

**Keywords:** high-speed shear wet granulation, agitation power consumption, monitoring by in-line near-infrared spectroscopy, partial least-squares regression, regression vectors, particle aggregation

## Abstract

The granulation process of pharmaceutical standard formulation in a high-speed shear wet granulation (HSWG) was measured by in-line near-infrared spectroscopy (NIRS) and agitation power consumption (APC) methods. The F-1, F-2, and F-3 formulations (500 g) contained 96% *w*/*w* α-lactose monohydrate (LA), potato starch (PS), and a LA:PS = 7:3 mixture, respectively, and all the formulations contained 4% *w*/*w* hydroxypropyl cellulose. While adding purified water at 10 mL/min, the sample powder was mixed. The calibration models to measure the amount of binding water (Wa) and APC of the HSWG formulations were established based on NIRS of the samples measured for 60 min by partial least-squares regression analysis (PLS). Molecular interaction related to APC between the particle surface and binding liquor was analyzed based on NIRS. The predicted values of Wa and APC for all formulations were superimposed with the measured values on a straight line, respectively. The regression vector (RV) of the calibration model for Wa indicated the chemical information of all the water in the samples. In contrast, the RV for APC suggested that APC changes in the processes are related to powder aggregation because of surface tension of binding water between particles.

## 1. Introduction

High-speed shear wet granulation (HSWG) is a widely used method as raw granules manufacturing process and is an important manufacturing process for solid oral dosage forms such as tablets in the pharmaceutical industry [1,2]. Because tableting raw granules with good flowability and tablet-ability for high-speed tableting could be prepared by using HSWG [1]. The bonding liquid is poured or sprayed into the mixed raw material powder bed and is stirred by the impeller agitates in the HSWG, and the surface tension of the liquid acts between the powder particles to form granules [3]. Many process parameters, such as the speed of rotation of the impeller and chopper, the amount of binder and the timing of its addition, and the agitating time of the wet mass, affect the formulation properties of the pharmaceutical preparations [4]. Previous studies [5,6,7] suggested that the added water amount during HSWG granulation was the most important factor affecting granule and tablet properties. The added binding water amount during HSWG was the most important factor affecting pharmaceutical properties as reported previously [5,6,7]. By measuring agitation power consumption (APC), torque, and temperature rise of the wet-mass during HSWG, the aggregation process was analyzed [8,9,10]. A study was reported to evaluate the properties of granulation from the measurements of power consumption and temperature of the HSWG [11]. They indicated that the granular properties can be controlled by changes in the APC curve of the granulation process in HSWG. The rate-controlling of the adding water amount in the mixed powder in HSWG was important to prepare high-quality granules, and the process could be monitored by the time–APC curves. The molecular mechanism related to binding water distribution between the powder particles during the HSWG for process optimization of the pharmaceutical properties of the formulation is significantly important. Process analysis technology (PAT) guidelines based on the quality-by-design concept were issued by the Food and Drug Administration in 2004 [12,13], and large-scale manufacturing of high-quality products using the PAT approach based on various kinds of analytical methods has been a significant challenge for the pharmaceutical industry. Raman spectroscopy can measure any liquid and solid sample [14], but not the inside of a powder sample because it measures the surface of the sample. Infrared spectroscopy can measure any sample with high accuracy [15], but it requires the sample to be diluted so that it’s use examples are limited. On the other hand, near-Infrared Spectroscopy (NIRS) is a useful tool for monitoring manufacturing processes in the pharmaceutical industry because it allows rapid and non-invasive measurement of changes in the physical and chemical properties of raw materials during the drug manufacturing process. Therefore, many NIRS studies were reported on the pharmaceutical properties of products during various processes [16,17,18,19,20,21,22]. In the early formulation development stage, NIRS was applied to predict the optimal water content for wet granulation [23,24,25,26]. Some of the studies [16,17,18,19,20,21,22] were monitoring the product quality during the manufacturing process, but they did not mention the actual molecular mechanism during the granulation process. In the previous study [27], to clarify the intrinsic granulation mechanism of HSWG, the dynamic granulation process of the simplest formulation based on glass beads was measured simultaneously by inline NIRS and APC methods. This new combination technique was used to clarify the molecular-level role, such as surface tension of water in the binding solution during the HSWG process, and the intermolecular interactions between powder particles and kneading liquid were analyzed based on both NIRS and APC data by multivariate regression analysis. In this study, a new analysis method by combining NIRS and APC was applied to the analysis of the molecular interaction of powder particles and binding water during the HSWG granulation process (Figure 1) for standard tablet formulation additive consisting of water-soluble lactose and water-swellable starch.

## 2. Results and Discussion

### 2.1. Change in APC-Time Profiles of All the Formulations during HSWG Processes

In the present study, to clarify the intermolecular interaction between particle and binding water during the HSWG process, the experimental condition was conducted far exceeding the ratio of powder and bound water comparing with real granulation as a model experiment.

Figure 2 shows the change in APC–time profiles in the HSWG process of each formulation sample (Table 1). In the granulation process of the wet powder mass (closed circle) based on F-1 consisting of α-Lactose monohydrate (LA), APC representing powder mixing resistance increased rapidly with increasing the amount of binding water (Wa) and reached the maximum APC value at a mixing time of 12 min (Wa = 90 mL). After that, it decreased as the Wa increased. In contrast, in the F-3 formulation consisting of potato starch (PS; closed triangle), the Wa was added dropwise until a mixing time of 22 min, but APC did not increase significantly. However, after 22 min of mixing (Wa = 190 mL), APC increased rapidly, and the maximum APC value was reached at 25 min. After that, the APC decreased as the Wa increased further. In the process based on F-2 consisting of a mixture of LA and PS (open circle), APC increased with increasing Wa, and it reached the maximum APC value at 17 min (140 mL); it then decreased as the Wa increased. The result of APC–time profiles of each sample showed markedly different variations in the lag time of the profiles, depending on the composition of the sample powder.

### 2.2. Change in NIRS Spectra of Various Formulations during HSWG Processes

To monitor the dynamic change in pharmaceutical properties based on intermolecular actions in the HSWG, NIRS spectra were measured during the processes. Figure 3 shows NIRS spectral changes in the model powder formulations during wet granulation processes in the HSWG. In the process of the F-1 formulation (Figure 3a, left), the spectral baseline was rapidly raised with increasing Wa, and the highest baseline was observed at 12 min. After the highest point, the baseline was reduced with the additional water and the increasing of peaks associated with water.

In the granulation of F-2, as the standard formulation mixture (Figure 3a, middle), the spectral baseline increased gradually with increasing Wa, and reached its peak at around 17 min after 150 mL of water was added. After the highest point, the baseline declined with additional Wa, but the increasing intensity of the peaks associated with water at 5250 and 6800 cm^−1^ was observed.

In contrast, in the process of the F-3 formulation (Figure 3a, right), the baseline was not significantly changed up to 20 min with increasing Wa, but after it increased rapidly and the highest baseline was observed at 23–25 min. After the highest value, similar to the other cases, the baseline declined and the peaks attributable to water increased.

In the previous granulation study to predict granular size by off-line NIRS [28], the 1st principle component had a plateau loading vector, suggesting that the granular size was tightly related with a baseline shift in the NIRS of aggregated wetting mass powder. Additionally, in the prediction study for particle size (50–1000 µm) of the bulk powder [29], NIRS light scattering dependent on the particle size of the bulk powder was related to baseline shifting of the diffused reflectance spectra, as expressed using Kubeluca–Munk equation.

### 2.3. Prediction of the Wa and the APC Values by Partial Least-Squares Regression (PLSR) Calibration Models

Multivariate analysis can quantitatively extract feature and regularity of large numerical data and it is effective for modeling of big data. In the present study, PLSR calibration models [30] were constructed to analyze the dynamic molecular mechanism of granulation based on second derivative NIRS spectra (explanatory variables) during the HSWG process and Wa and APC (objective variables). The calibration models for the Wa and APC were selected to minimize the SECV by the leave-one-out method in the PLSR, respectively.

Figure 4 shows the plots between actual and predicted values of Wa and APC of the HSWG process by using NIRS. The plots in all the formulations gave a straight line with a correlation coefficient constant more than (r) = 0.98, with slopes close to 1. Table 2 shows the chemometric parameters of the fitted models to predict the Wa and APC values, such as the cumulative percent variance (CPV), minimized the standard error of cross-validation (SEV), standard error of calibration (SEC), the prediction residual error sum of squares (PRESS) for calibration and validation (PRESS Cal and PRESS Val), and r-values for calibration and validation (r-Cal and r-Val). The PLSR calibration models for the Wa values in all the formulations consisted of two latent variables (LV). The models for the APC values in the F-1 and F-2 formulations also consisted of two LVs, but that of the F-3 formulation consisted of three LVs. The results (Figure 4 and Table 2) indicated that the fitted PLSR calibration models show a sufficient linear relationship between actual and predicted values of the Wa and APC values. The predicted APC values during the HSWG processes using NIRS are represented in Figure 2 as dotted lines, and the APC-time profiles superimposed well with the measured APC values in all the formulations.

### 2.4. Quantitative Relationships between Wa and NIRS Spectra of Wet Masses in the Formulation Consisting of LA and/or PS during HSWG Processes

Figure 5A–C and Figure 6A–C show the LVs and the scores of the calibration models to measure Wa in all the model powder formulations. Figure 7 shows the second derivative NIRS spectra for the band 5000–5500 cm^−1^ of all the formulations during HSWG processes.

In the F-1 formulation (LA), LV1 and LV2 in Figure 5A showed major peaks at 5245 cm^−1^ and 5168 cm^−1^, respectively, for which the percent variabilities (PVs) were 95.5% and 2.6% because the LV1 and LV2 were attributable to free water and crystal water of monohydrate crystalline LA, respectively [31]. The score of LV1 (Free water) for F-1 (Figure 6A) increased with increasing mixing time from 2 min to 12 min, it was sustained at 12–23 min, and then it increased again at 23–52 min. The score of LV2 (Crystal water) decreased with increasing mixing time from 2 min to 12 min, because LA dissolved, and it was almost constant at 12–52 min. Here, because LA is a soluble material (solubility of LA monohydrate in water is 16 g/100 mL at 20 °C) [32], more than 17% *w*/*w* of crystalline LA monohydrate was dissolved in the binding water, meaning that the peak at 5168 cm^−1^ attributable to crystalline LA decreased during granulation. The LVs and the scores for F-1 were reflected in the second derivative NIRS spectra (Figure 7A), the negative peak at 5249 cm^−1^ increased with increasing mixing time, but that at 5168 cm^−1^ decreased in the initial mixing period at 2–12 min. So, the changes in both peaks at 5249 and 5168 cm^−1^ were due to free water and crystal water, respectively.

Focusing on the second derivative NIRS spectral change (Figure 7A), the negative peak at 5249 cm^−1^ increased with the addition of water, but the negative peak at 5168 cm^−1^ decreased gradually when 100 mL of water was added up to 12 min, and after that, the peaks increased sharply. The peak change at 5332 cm^−1^ (Figure 3a and Figure 7A) might be due to a baseline shift. LA monohydrate dissolves 17% *w*/*w* in binding solution, so the decrease in the peak at 5168 cm^−1^ up to 12 min was due to their dissolution. The result suggested that there was a tight relationship between the change in all peaks and the change in Wa.

In the F-2 formulation (Mixture), the LV1 and LV2 (Figure 5B) showed negative peaks at 5245 cm^−1^ and 5168 cm^−1^, for which the PVs were 90.5% and 1.2%, respectively, because the peaks were due to free water and crystal water, respectively, which was similar to F-1. The score of LV1 (Free water) of F-2 (Figure 6B) increased with increasing mixing time at 2–52 min, and the score of LV2 (Crystal water) was almost the constant at 2–52 min, the same as those of F-1. Both the profiles of the scores of LV1 and LV2 were intermediate between those of F-1 and F-3, respectively. In Figure 7B, the peak at 5249 cm^−1^ increased, but that at 5168 cm^−1^ decreased at 2–17 min, and it increased at 17–52 min.

In contrast, in the F-3 formulation (PS), the LV1 and LV2 in Figure 5C showed negative peaks at 5245 and 5180 cm^−1^, in which the PVs were 95.7% and 0.8%, respectively. The peak at 5245 cm^−1^ was attributable to free water, and a broad peak was observed at 5180 cm^−1^ attributable to water that interacted with PS, as with the intermediate peak between crystalline and free water [31]. In Figure 6C, the score of LV1 (Free water) for F-3 increased constantly with increasing mixing time of 2–52 min. In contrast, the score of LV2 (Absorbed water) had some peaks at 8–27 min but did not increase significantly. The result reflected on the second derivative spectra of the F-3 (Figure 7C), the negative peak at 5180 cm^−1^ increased with increasing 2–10 min mixing time then it decreased, and shifted to 5249 cm^−1^ at 10–25 min, then it increased again after 23 min. In the initial stage of mixing time at 2–10 min, the PS absorbed water and swollen, so the peak at 5180 cm^−1^ increased.

In general, it is well known that red and blue shifts on infrared spectra [33] are popular phenomena related to crystalline transformation caused by molecular interaction. Zelent at al. [34] concluded that freezing shifts the absorption of the stretch mode lower (redshift) based on the infrared spectra of ice and liquid water. In our previous report [35], it was demonstrated that theophylline anhydrate transformed into its monohydrate during granulation by added binding water, and then, peak at 5160 cm^−1^ (added free water) shift to 5070 cm^−1^ (crystalline water). The spectral information indicated that free water was bound by intermolecular interactions with surrounding molecules and the OH group peaks were red-shifted. In this study, the absorption peaks of 5322, 5207, and 5168 cm^−1^ were assigned to free water, bound water, and crystalline water, respectively, and the reports of the redshift due to the intermolecular interaction of the OH group were theoretically consisting with our results.

### 2.5. Quantitative Relationships between APC and NIRS Spectra of Wet Masses of the Formulations Consisting of LA and/or PS during HSWG Processes

To clarify the molecular level granulation mechanism of the HSWG process, the relationships between the APC and NIRS spectral change were characterized using the PLSR method. Figure 5D–F and Figure 6D–F show the LVs and scores of the calibration models to measure the APC of all model powder formulations.

In the case of the F-1 formulation consisted of LA (Figure 5D), the calibration models consisted of LV1 and LV2 with high contribution ratios PVs of 48.6% and 44.1%, respectively. In the LV1, there were negative and positive peaks at 5322 and 5245 cm^−1^, respectively, attributable to baseline shifting and bonding water. In the LV2, there was a negative peak at 5168 cm^−1^ attributable to the crystal water of LA. In Figure 6D, the score profile for the LV1 increased rapidly up to 12 min, and after it decreased, so the profile was very similar to the APC profile in Figure 2. In contrast, the score for the LV2 decreased and increased at 2–15 min related to dissolution/hydration of LA, and it reached a minimum at 17 min, and then it increased.

The peak change at 5332 cm^−1^ (Figure 3a and Figure 7A) might be due to a baseline shift. The result suggested that there was a tight relationship between the change in all the peaks and the change in APC related to viscous resistance. Because, there was a high solid–liquid interaction between water and the surface of LA particles, which has crystal water on a surface, and it has surface has crystal water on the surface, and so these might form an interaction between the crystal water and free water. In other words, the attractive force between water molecules and LA molecules is weak, but the attraction between the water molecules works strongly. LA monohydrate dissolves 17% *w*/*w*, so the decrease in the peak at 5168 cm^−1^ up to 12 min was due to their dissolution. Viscous drag was after a maximum at 12 min of mixing time, because the surface tension of the whole wet mass was at a maximum, and it means that the liquid phase between LA particles formed a liquid bridge, as shown in the Funicular or Capillary model in Figure 8. It was thought that the liquid bridges between particles disappeared and formed a slurry by adding further water, and their viscous drag decreased significantly.

In the F-2 formulation (the mixture), as shown in Figure 5E, the calibration models to measure APC consisted of LV1 and LV2, and their PVs were 29.8% and 18.3%, respectively. The LV1 had negative and positive peaks at 5322 and 5245 cm^−1^, respectively, attributable to the baseline shift and free water, and the LV2 had positive and negative peaks at 5253 and 5168 cm^−1^, respectively, attributable to free water and crystal water. In Figure 6E, the score LV1 profile of F-2 increased up to 17 min, then it decreased. The score for LV2 decreased and increased at 2–15 min related to the dissolution/hydration of LA, and it showed a minimum at 17 min, then it increased. The score LV1 profile of F-2 for APC were almost similar to that of LA and highest at 17 min.

In the F-3 formulation consisting of PS, as shown in Figure 5F, the calibration model to measure APC consisted of LV1 and LV2 with high contribution ratios (i.e., PVs) of 31.9% and 32.6%, respectively. The LV1 had positive and negative peaks at 5361 and 5257 cm^−1^, respectively, attributable to a baseline shift and free water, and the LV2 had positive and negative peaks at 5291 and 5207 cm^−1^, respectively, attributable to free water and absorbed water. In Figure 6F, the score profile for LV1 had minimum and maximum peaks at 17 and 27 min. The scoring profile (Absorbed water) for LV2 had maximum and minimum peaks at 17 and 27 min. The scoring profile of LV 1 showed almost the same pattern as that of LV2, but it was upside down.

### 2.6. Molecular Level Granulation Mechanism during HSWG Processes of the Wet Mass Formulations Consisting of LA and/or PS Based on Changes in APC Values and NIRS

PLSR is effective for the extraction of features and regularity and the modeling of large numerical data. PLSR also can construct best-fitted calibration models with a linear relationship between explanatory variables and objective functions. Therefore, the regression vectors (RVs) of the best-fitted models can explain and reflect the chemical background of the models. As shown in Figure 4, there were sufficient linear relationships between the predicted and measured values of the Wa or APC for all formulations during HSWG processes, and their RVs are shown in Figure 9.

First, the mechanisms to measure Wa in standard powder formulations consisting of lactose and starch were considered based on the individual RVs obtained using NIRS. The RV of the calibration model for Wa of the F-1 formulation, which involved a CPV of 98.2%, had negative peaks at 5168 and 5261 cm^−1^ and a positive peak at 5326 cm^−1^. Therefore, in the second derivative spectra (Figure 7A), the peak at 5168 cm^−1^ related to crystal water of LA monohydrate decreased by dissolution, but the peak at 5261 cm^−1^ increased due to the formation of aggregates by adding water. In contrast, in the RV of the F-3 formulation, which involved a CPV of 96.5%, a negative peak at 5249 cm^−1^ related to free water and a positive peak at 5300 cm^−1^ were observed. The negative peak at 5249 cm^−1^ increased with increasing amounts of added water, and the peak at 5261 cm^−1^ increased due to the formation of aggregates by adding water, as shown in Figure 7C. The RV peak pattern of the F-2 formulation (mixture), which involved a CPV of 91.7%, was intermediate between those of the F-1 and F-3 formulations because F-2 contained both LA and PS. The peaks of the RVs of F-1, F-2, and F-3 reflected the peak changes of the individual second derivative NIRS due to increasing amounts of added water in Figure 7A–C, respectively.

Second, the molecular mechanisms to measure APC during HSWG processes of formulations consisting of LA and PS were clarified based on the individual RVs obtained by NIRS. In the RVs of the calibration models for APC (Figure 9B), the model of the F-1 formulation involved a CPV of 92.8%, and a negative peak at 5318 cm^−1^ and a positive peak at 5249 cm^−1^ were observed in the RV. The RV for F-2 had also almost the same peak pattern as for F-1, but the CPV was much less (48.1%) than that of F-1. The model of F-3 involved a CPV of 68.7%, and a negative peak at 5249 cm^−1^ and a positive peak at 5322 cm^−1^ were observed in the RV, but the peak RV pattern of F-3 was upside down compared with that of F-1 or F-2. The peak at 5249 cm^−1^ on RV may be attributed to some of the free water used to bind the powder particles. Because it is well known that the surface tension of free water between particles [11], this is utilized to make granules as a binding force for the particles in wet granulation (Figure 8). The peak at 5322 cm^−1^ may be related to the aggregation of particles due to the baseline shift shown in Figure 3b. The result indicated that APC change during HSWG processes are related to particle aggregation due to surface tension between the powder particles.

All the results of the RVs indicated that the calibration models to measure Wa are significantly different from those for APC. Because Wa measured the total water amount involving free water, bonding water, and crystal water in a powder wet mass, but APC measured the total surface tension induced by part of the water between powder particles in powder aggregates.

On the other hand, the result of the RVs of the calibration models in the above section were indicated as follows; although both models were calculated based on the same NIRS data sets, the RVs were remarkably different, because the information extracted from the basic data sets was different depending on the objective variable. Therefore, specific NIRS spectra to measure Wa or APC values can be reconstructed from respective calibration model parameters by an inverse calculation.

Figure 10A–C shows the reconstructed NIRS spectra of F-1, F-2, and F-3, respectively, calculated based on the calibration model parameters to measure Wa. The reconstructed spectra were almost the same profiles as those of the second derivative spectra, as shown in Figure 7A–C, because the calibration models measured the total amounts of all kind water involving crystal water, bonding water, and free water in all the formulations, respectively.

In contrast, Figure 10D–F shows the reconstructed spectra of F-1, F-2, and F-3, respectively, based on the model parameters to measure APC values. The reconstructed spectra of the model for APC were significantly different from the original spectra (Figure 7A–C). Additionally, the reconstructed spectra obtained based on APC values were significantly different from those based on Wa (Figure 10). The difference indicated that the calibration models to measure APC were involved in the extracted information of limited water-related to surface tension between powder particles in the wet masses of all the formulations. All of the NIRS spectral information reflected the roles of water in all formulations during the HSWG processes, so the particle agglomeration models of the LA and PS formulations were summarized, and shown in Figure 8.

LA particles were partially and rapidly dissolved in binding water in the initial stage of granulation, then binding water bridges were formed between LA particles by a limited amount of water (surface water), which was thought to increase viscous resistance by surface tension. Because the particle size of LA is 200 µm and that of PS is around 50 µm, and also the density of PS is lower than that of LA, when comparing LA with the same mass of PS, PS has a larger a total surface area than LA. In the case of the PS formulation, PS absorbed water for the swelling at initial granulation stage, so the water amount to use for the granulation of PS in F-2 and F-3 decreased, it was considered that the amount of water necessary for maximizing viscous resistance of PS was larger than that of LA. Therefore, the staring time of granulation in F-2 and F-3 was delay depended on the amount of PC. In all granulation cases, as shown in Figure 2 and Figure 8, after maximizing viscous resistance during HSWG processes, APC decreased rapidly by making a slurry with additional water. This meant that powder particles were suspended into the binding water (Figure 8, upper), so binding forces between the powder particles formed due to surface tension of the binding liquor disappeared, as shown in Figure 2.

## 3. Materials and Methods

### 3.1. Materials

LA (average particle size was 200 µm; Lot No.3033 2177 43129M), included as a diluent, was obtained from DFE Pharma (Corporate Head Office, Goch, Germany). PS (average particle size was 50 µm; Lot No.TC-10), used as the disintegration agent, was obtained from Kosakai Pharmaceutical Co., Ltd. (Nagaoka, Japan). Hydroxypropyl cellulose (HPC-L; Lot No. NEA-4131), as the binder agent, was obtained from Nippon Soda Co., Ltd. (Tokyo, Japan). As shown in Table 1, the LA/PS standard formulation powder [30] contained LA and PS at a ratio of 7:3, and the raw powder sample was added and mixed with 4% *w*/*w* of HPC-L as a binder with 96% *w*/*w* of the standard formulation powder.

### 3.2. Methods

#### 3.2.1. Granulation

The outline of the HSWG with the NIRS monitoring system is shown in Figure 1. The HSWG (FS 2 type high speed mixer, internal volume 2.0 L, Earth Technica Co., Ltd., Tokyo, Japan) with a power measurement device for stirring and NIRS diffuser and the injector of a binder solution was used for granulation. The sample powder (total weight 500 g, Table 1) was placed in a chamber and mixed by the impeller at 500 rpm and the cross-screw at 1000 rpm for 2 min. Then, without stopping the apparatus, purified water (total 500 mL) was added dropwise at an addition rate of 10 mL/min for 50 min with a peripheral pump, and stirring was continued for 8 min, and then the sample powder was mixed for a total of 60 min. The analog signals of the APC output from the HSWG granulator were recorded using a personal computer every minute through an A/D converter. Stirring power consumption was calculated based on the converted numerical values using the following equation.
APC(W)=(V×480)/5

Here V is voltage. The APC data were measured at a sampling rate of 10 data/s for 60 min (36,000 data), and the values were averaged for each minute.

#### 3.2.2. Measurement of NIRS

The NIRS were measured from 4000 to 12,500 cm^−1^ with 8 cm^–1^ resolution, 118 scans using an NIRS instrument (MPA, Bruker Optics, Germany) every min for 60 min with a diffused fiber-optic probe during the HSWG process.

#### 3.2.3. Calibration Models to Predict the Pharmaceutical Properties in the Granular Formulated Powder Mixture

The calibration models to predict the Wa and the APC during the granulation were obtained as follows: the NIRS spectra were measured at every min for 60 min during adding binding water, and a total of 60 spectra were obtained for calibration. After NIRS spectra were pre-treated using a second derivative function, the calibration models were determined the Wa and APC as objective functions by the leave-one-out method in PLSR analysis (Pirouette Ver. 4.5, Infometrix Co., Woodenville, WA, USA) [36]. The CPV, SEV, SEC, PRESS Cal, PRESS Val, r-Cal, and r-Val were obtained as chemometric parameters.

## 4. Conclusions

To clarify the dynamic molecular mechanism of the wet-granulation of powder particles of standard formulations containing LA/PS/HPC in HSWG processes, the process big data (APC and NIRS) of wet granulation were measured and analyzed using PLSR. The best-fitted PLSR calibration models of all the formulations were obtained based on NIRS spectra as the explanatory variable, and Wa or APC as the objective variables during HSWG processes. The RVs to measure Wa were significantly different from those for APC, because RVs for Wa are to measure the total water amount involving free water, bonding water, and crystal water in a powder wet mass, but that for APC was to measure total surface tension induced by water between the powder particles in powder aggregates. The dynamic molecular mechanism of HSWG was indicated by NIR, as follows: LA particles were partially and rapidly dissolved in binding water in the initial stage of granulation, after then binding water bridges were formed granules by surface tension. In contrast, PS absorbed water for the swelling at the initial granulation stage, so the water amount to use for the granulation decreased, the staring time of granulation was delay depended on the amount of PC.

## Figures and Tables

**Figure 1 pharmaceuticals-13-00226-f001:**
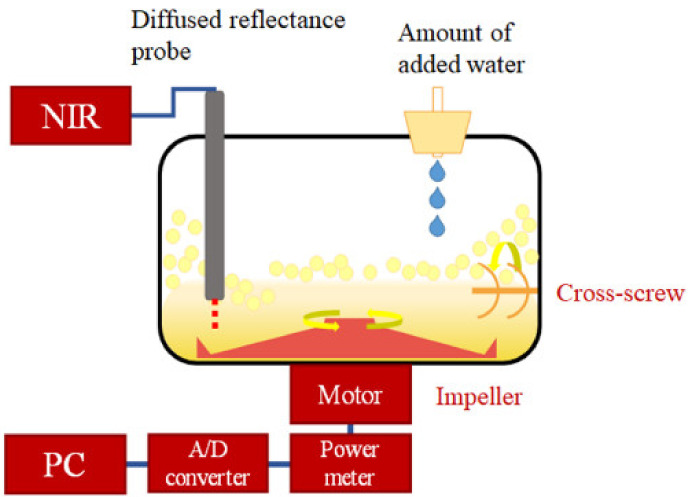
Schematic diagram of the high-speed stirring granulator with the near-infrared spectral monitoring system.

**Figure 2 pharmaceuticals-13-00226-f002:**
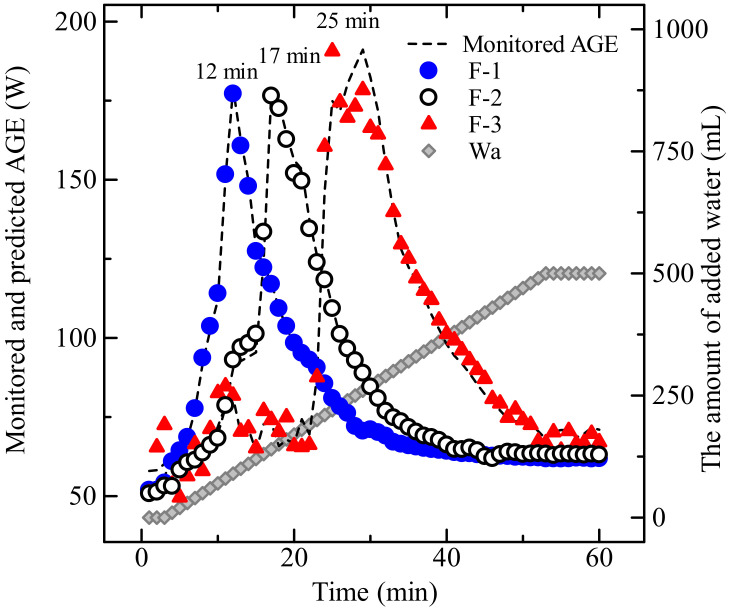
Changes in agitation power consumption–time profiles in the granulation processes of the model powder formulations.

**Figure 3 pharmaceuticals-13-00226-f003:**
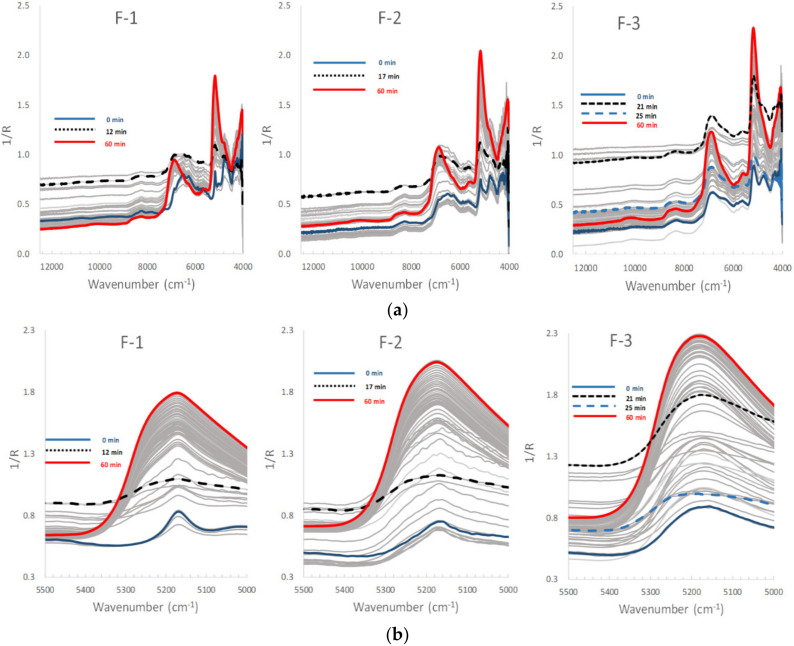
(**a**) Near-infrared spectral changes in the model powder formulations wet granulation processes using the high-speed shear wet granulator. (**b**) Near-infrared spectral changes (5500–5000 cm^−1^) in the model powder formulations during wet granulation processes using the high-speed shear wet granulator.

**Figure 4 pharmaceuticals-13-00226-f004:**
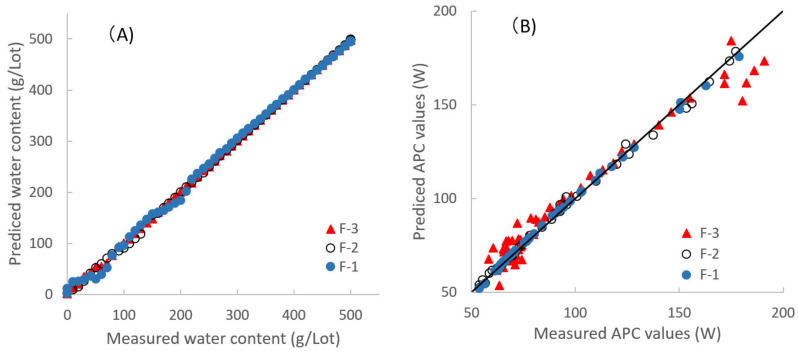
Relationships between measured values and predicted values of the amount of water added and the agitation power consumption during high-speed shear wet granulation processes. (**A**) The amount of water added (Wa); (**B**) the agitation power consumption (APC).

**Figure 5 pharmaceuticals-13-00226-f005:**
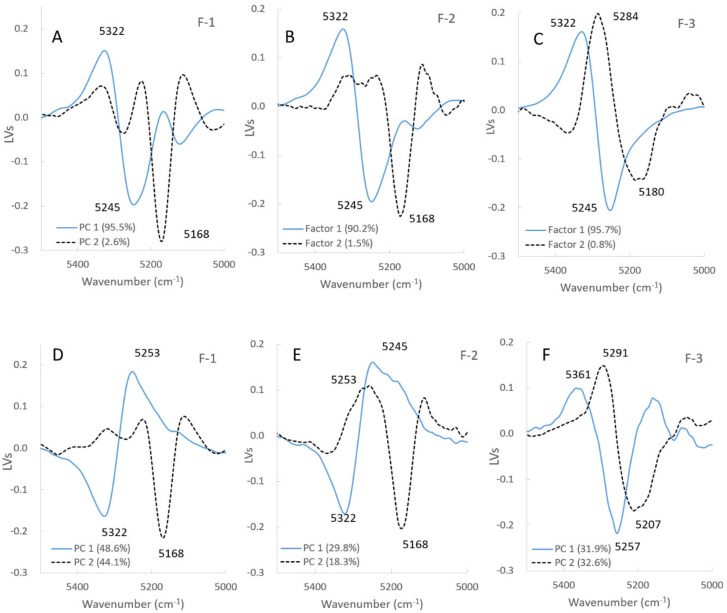
(**A**–**C**) Latent variables of the calibration models to predict the amount of water added and the agitation power consumption of all formulations. (**A**–**C**), the amount of water for lactose, mixture, and starch samples. The solid line is LV1, the dot line is LV2. (**D**–**F**) Latent variables of the calibration models to predict the amount of water added and the agitation power consumption of all formulations. (**D**–**F**), the agnation power consumption for lactose, mixture, and starch samples. The solid line is LV1, dot line is LV2.

**Figure 6 pharmaceuticals-13-00226-f006:**
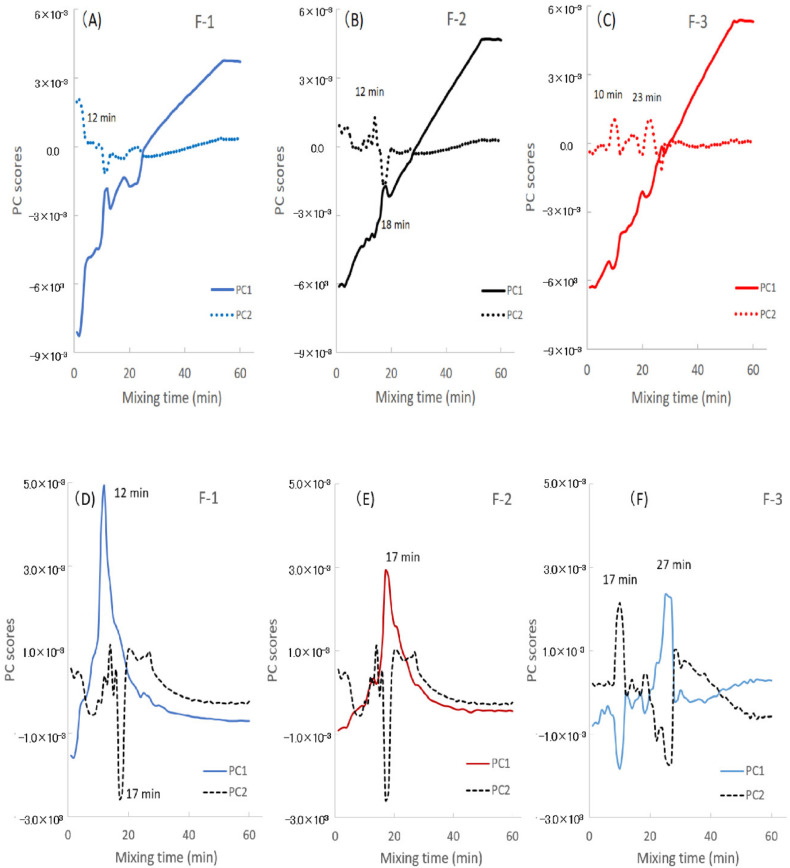
(**A**–**C**) Score–time profiles of the partial least-squares regression calibration models for the amount of water added and the agitation power consumption during high-speed shear wet granulation processes of all formulations. (**A**–**C**), the amount of water for lactose, mixture, and starch samples. The solid line is LV1, the dot line is LV2. (**D**–**F**) Score–time profiles of the partial least-squares regression calibration models for the amount of water added and the agitation power consumption during high-speed shear wet granulation processes of all formulations. (**D**–**F**), the agnation power consumption for lactose, mixture, and starch samples. The solid line is LV1, the dot line is LV2.

**Figure 7 pharmaceuticals-13-00226-f007:**
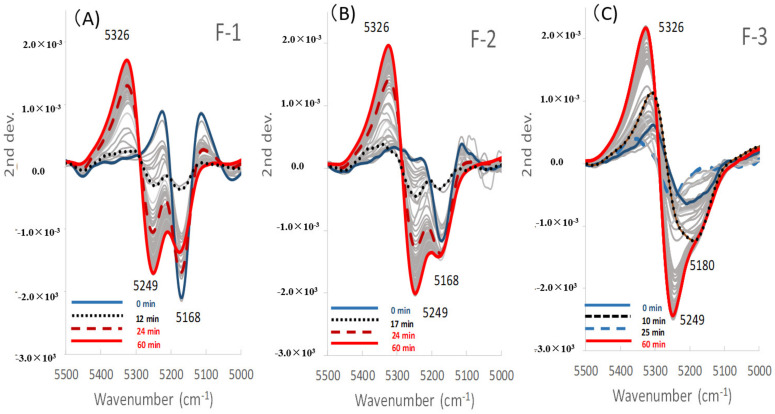
Second derivative near-infrared spectra for the band 5000–5500 cm^−1^ of the model powder formulations during the high-speed shear wet granulation processes. (**A**) The lactose sample, (**B**) mixture sample, and (**C**) starch sample.

**Figure 8 pharmaceuticals-13-00226-f008:**
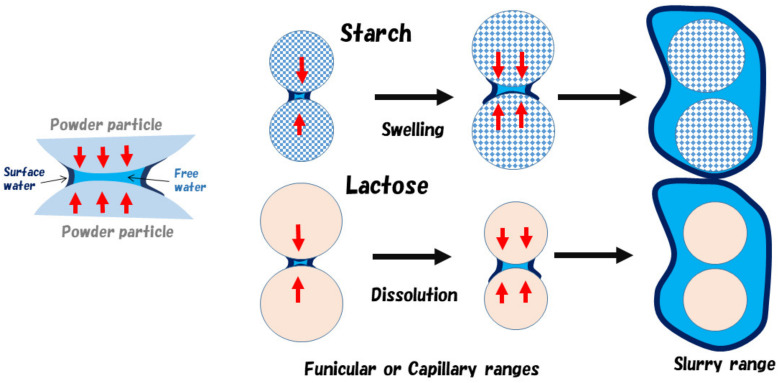
Schematic diagrams of water bridges between the formulation particles in granules during high-speed shear wet granulation processes.

**Figure 9 pharmaceuticals-13-00226-f009:**
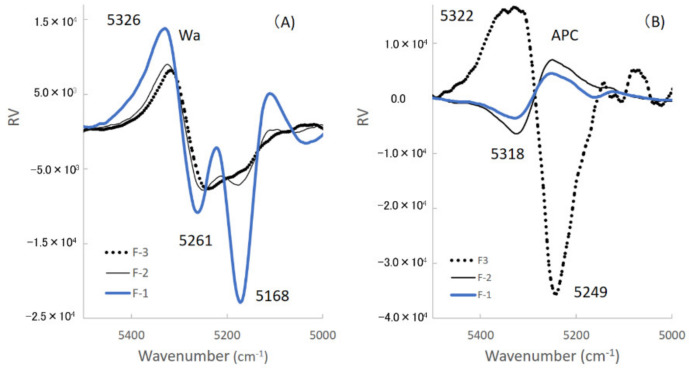
Regression vectors of the calibration models to predict the amount of water added and the agitation power consumption respectively of all formulations. (**A**) The amount of water added (Wa); (**B**) the agitation power consumption (APC).

**Figure 10 pharmaceuticals-13-00226-f010:**
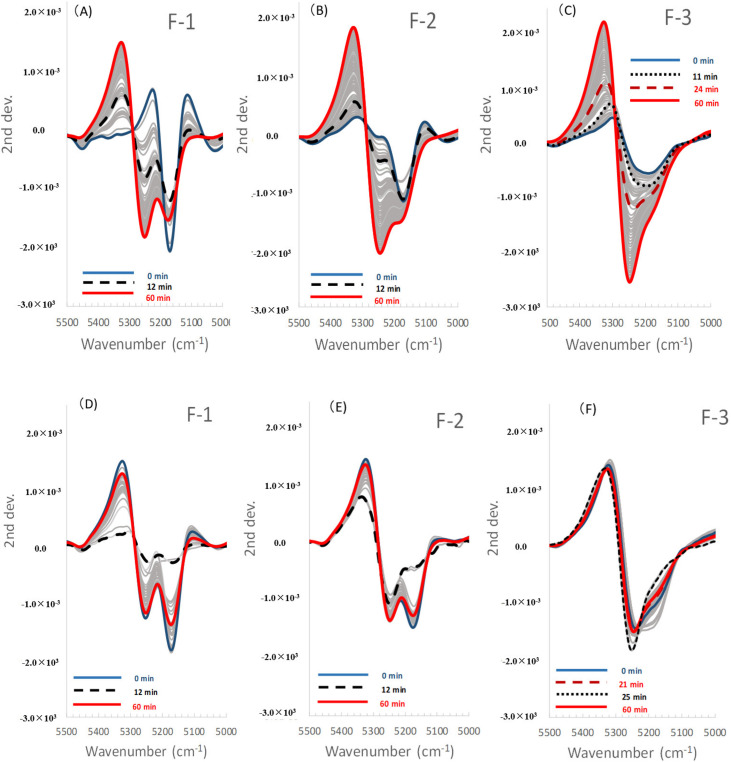
(**A**–**C**) Reconstructed near-infrared spectra for the band 5000–5500 cm^−1^ obtained based on chemometric parameters of the best-fitted calibration models for all formulations. (**A**–**C**), the amount of water for lactose, mixture, and starch samples. (**D**–**F**) Reconstructed near-infrared spectra for the band 5000–5500 cm^−1^ obtained based on chemometric parameters of the best-fitted calibration models for all formulations. (**D**–**F**), the agnation power consumption for lactose, mixture, and starch samples.

**Table 1 pharmaceuticals-13-00226-t001:** Compositions of the standard powder formulations consisting of lactose and starch.

Matreials	F-1		F-2		F-3	
	(g)	(%)	(g)	(%)	(g)	(%)
LA	480	96	336	67.2	0	0
PS	0	0	144	28.8	480	96
HPC-L	20	4	20	4	20	4
Wa	500	100	500	100	500	100

LA, α-lactose monohydrate; PS, potato starch, HPC-L, hydroxypropyl cellulose-L; Wa, purified water.

**Table 2 pharmaceuticals-13-00226-t002:** The chemometric parameters for the fitted calibration models to measure the added water amount and the agitation power consumption values.

Wa	PC	CPV	SEV g/lot	Press Val	r Val	SEC g/lot	Press Cal	r Cal
F-1	2	98.2	7.10	3.03×10^3^	0.999	7.14	2.90×10^3^	0.999
F-2	2	91.7	3.37	6.79×10^2^	1.000	2.22	2.82×10^2^	1.000
F-3	2	96.5	2.09	32.62×10^2^	1.00	1.84	1.92×10^2^	1.000
**APC**	**PC**	**CPV**	**SEV, W**	**Press Val**	**r Val**	**SEC, W**	**Press Cal**	**r Cal**
F-1	2	92.8	1.09	37.17×10	0.999	1.01	5.81×10	0.999
F-2	2	48.1	1.74	1.82×10^2^	0.999	2.14	2.61×10^2^	0.998
F-3	3	68.7	7.92	3.77×10^3^	0.981	6.64	2.47×10^3^	0.986

CPV, cumulative percent variance; SEV, standard error of cross-validation; SEC, standard error of calibration; PRESS Cal and PRESS Val, the prediction residual error sum of squares for calibration and validation; r-Cal and r-Val values, correlation coefficient constant for calibration and validation.

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
