# Peer review of "Elucidation of the Molecular Mechanism of Wet Granulation for Pharmaceutical Standard Formulations in a High-Speed Shear Mixer Using Near-Infrared Spectroscopy"

_pharmaceuticals, 2020, doi:10.3390/ph13090226_

Round 1

Reviewer 1 Report

None

Author Response

Thank you very much for your comments.

Reviewer 2 Report

I enjoyed reading the manuscript entitled ‘Elucidation of the molecular mechanism of wet granulation for pharmaceutical standard formulations in a high-speed shear mixer using near-infrared spectroscopy’ and the results are interesting. I would recommend this manuscript to be published after a minor revision for the following comments.

  • Add a space between the numbers and units.
  • Introduction: Add the details of different analytical tools used in the literature for monitoring the manufacturing process and justify the selection of NIR technique compared to the others.
  • Table 1: Correct the purified water row of Table 1.
  • Figure 2: Correct the font case of the labels and space before the units. Correct the data labels and present them as F-1, F-2 and F3.
  • Figures needs to be updated with a clear text.
  • Page 2, Section 2.1: What do you mean by ‘pure water’. Is binding water and pure water are different. Keep the words consistent.
  • Page 2, Section 2.1: The formulations F-2 and F3 discussion not matching with the Table 1. According to the Table 1, F-2 consists of potato starch and F-3 consists of a mixture. However, the discussion in Section 2.1 is incorrect. Correct.
  • Page 2, Section 2.1, line 8: Correct the volume of water. Is it 230 mL or 210 mL?
  • Page 3 and 4, Section 2.2.: Keep the discussion in an order of F-1, F-2 and F-3. It is confusing to follow. Change the order of figures to F-1, F-2 and F-3
  • Page 4, paragraph 3, line 4: correct ‘scatting’

Author Response

List of alternation.

Thank you very much for your comments and suggestions.

We were very carefully considered your comments and suggestions, and then corrected all of them everywhere in the text.

Author's Reply to the Review Report (Reviewer 2)

I enjoyed reading the manuscript entitled ‘Elucidation of the molecular mechanism of wet granulation for pharmaceutical standard formulations in a high-speed shear mixer using near-infrared spectroscopy’ and the results are interesting. I would recommend this manuscript to be published after a minor revision for the following comments.

  • Add a space between the numbers and units.

=====>>>Thanks. We added a space between the numbers and units in everywhere in the text.

  • Introduction: Add the details of different analytical tools used in the literature for monitoring the manufacturing process and justify the selection of NIR technique compared to the others.

=====>>>The sentences concerning to discussion different analytical tools were added in Introduction on page 2, line 12-18, as “Process analysis technology (PAT) guidelines based on the quality-by-design concept were issued by the Food and Drug Administration in 2004 [12-13], and large-scale manufacturing of high-quality products using the PAT approach based on various kinds of analytical methods has been a significant challenge for the pharmaceutical industry. Raman spectroscopy can measure any liquid and solid sample [14], but not the inside of a powder sample because it measures the surface of the sample. Infrared spectroscopy can measure any sample with high accuracy [15], but it requires the sample to be diluted, so that its use examples are limited. On the other hand, near-Infrared,,,,,,”.

  • Table 1: Correct the purified water row of Table 1.

=====>>>Table 1 was corrected following your suggestion.

  • Figure 2: Correct the font case of the labels and space before the units. Correct the data labels and present them as F-1, F-2 and F3.

=====>>>The labels and space before the units in Figure 2 were corrected following your suggestion.

  • Figures needs to be updated with a clear text.

=====>>>All of Figures were updated following your suggestion.

  • Page 2, Section 2.1: What do you mean by ‘pure water’. Is binding water and pure water are different. Keep the words consistent.

=====>>>"pure water" were replaced to ‘the Wa’ on page 2. line 44,45,48,50.

  • Page 2, Section 2.1: The formulations F-2 and F3 discussion not matching with the Table 1. According to the Table 1, F-2 consists of potato starch and F-3 consists of a mixture. However, the discussion in Section 2.1 is incorrect. Correct.

=====>>>In Table 1, F-2 and F-3 were wronged, so, Table 1 was corrected following your suggestion.

  • Page 2, Section 2.1, line 8: Correct the volume of water. Is it 230 mL or 210 mL?

=====>>>In Section 2.1, since the explanation concerning to the value of Wa in Section 2 was not correct, the W value at each measurement time was corrected to an accurate value so that the sentence would be an accurate expression, as shown in the section on page 2, line 44-50, as “a mixing time of 12 min (Wa=90 mL). ,,, after 22 min of mixing (Wa=190 mL), ,,,,,, APC value at 17 min (140 mL)”.

  • Page 3 and 4, Section 2.2.: Keep the discussion in an order of F-1, F-2 and F-3. It is confusing to follow. Change the order of figures to F-1, F-2 and F-3

=====>>>As you pointed out, the discussion flow in Section 2.2, Section 2.4 and Section 2.5 has been changed to F-1, F2, F3, on page 4, line 1-8, on page 6, line 30-47, on page 9, line 33 to page 10, 1-8.

  • Page 4, paragraph 3, line 4: correct ‘scatting’

=====>>>Miss spell was corrected to “scattering” in the text on page 4, line 12.

Reviewer 3 Report

This paper introduces near-infrared spectroscopy (NIR) to elucidate the molecular mechanism of wet granulation for pharmaceutical standard formulations during a high-speed shear mixer.  The major claim of this paper is that by introducing NIR to probe the chemical information of the water state in the powder bed. While the motivation is sound, the idea of using the PLS model to fit the signals with water amount and agitation power consumption is straightforward. I, therefore, have reservations about its novelty.

I also concern about peak assignments.  “LV1 and LV2 were attributable to free water and crystal water of monohydrate crystalline LA, respectively [29].” “The peak at 5245 cm-1 was attributable to free water, and a broad peak was observed at 5180 cm-1 attributable to water that interacted with PS, as with the intermediate peak between crystalline and free water methods [29].” I would like to see clear evidence.

I have another concern about the water amount used in this work.  The 1:1 ratio of solid to water is too low for real manufacturing.  

Author Response

Author's Reply to the Review Report (Reviewer 3)

Comments and Suggestions for Authors

This paper introduces near-infrared spectroscopy (NIR) to elucidate the molecular mechanism of wet granulation for pharmaceutical standard formulations during a high-speed shear mixer.  The major claim of this paper is that by introducing NIR to probe the chemical information of the water state in the powder bed. While the motivation is sound, the idea of using the PLS model to fit the signals with water amount and agitation power consumption is straightforward. I, therefore, have reservations about its novelty.

====>>>>Thanks for your nice comment. We believe that molecular interaction between powder and binding water during high speed granulation was not reported yet.

I also concern about peak assignments.  “LV1 and LV2 were attributable to free water and crystal water of monohydrate crystalline LA, respectively [29].” “The peak at 5245 cm-1 was attributable to free water, and a broad peak was observed at 5180 cm-1 attributable to water that interacted with PS, as with the intermediate peak between crystalline and free water methods [29].” I would like to see clear evidence.

====>>>> It is well known that red and blue shifts on IR spectra (Scheiner et al.) are popular phenomena related to crystalline transformation caused by molecular interaction. Zelent at al. concluded that based on infrared spectra of ice and liquid water, freezing shifts the absorption of the stretch mode lower (red shift). In our pervious report, it is demonstrated that theophylline anhydrate transformed into its monohydrate during granulation by added binding water, and then, peak at 5160 cm-1 (added free water) shift to 5070 cm-1 (crystalline water). The spectral information indicated that a free water was constrained by the intermolecular interaction with the surrounding molecules and the peak of the OH group was red-shifted. Therefore, the related sentences were corrected on page 6, line 48 to page 7, line 1-5, as “”.

====>>>>Additionally, the related references [33, 34, 35] were added in the text.

  1. Scheiner, S., & Kar, T. (2002). Red-versus blue-shifting hydrogen bonds: are there fundamental distinctions?. The Journal of Physical Chemistry A, 106(9), 1784-1789.
  2. B. Zelent and J. M. Vanderkooi, Infrared spectroscopy used to study ice formation: the effect of trehalose, maltose and glucose on melting, Anal Biochem. 2009 Jul 15; 390(2): 215–217. doi: 10.1016/j.ab.2009.04.019
  3. Otsuka, M., Kanai, Y., & Hattori, Y. (2014). Real‐Time Monitoring of Changes of Adsorbed and Crystalline Water Contents in Tablet Formulation Powder Containing Theophylline Anhydrate at Various Temperatures During Agitated Granulation by Near‐Infrared Spectroscopy. Journal of Pharmaceutical Sciences, 103(9), 2924-2936.

I have another concern about the water amount used in this work.  The 1:1 ratio of solid to water is too low for real manufacturing. 

====>>>>The 1:1 ratio of solid and water is useful exponential condition, in order to clarify the intermolecular interaction between particle and binding water during HSWG process, as a model experiment. Therefore, the related sentences were added in the text, page 2, line 37-40, as “In the present study, in order to clarify the intermolecular interaction between particle and binding water during HSWG process, experimental condition was conducted far exceeding the ratio of powder and bound water comparing with real granulation as a model experiment.”.

Additional correction.

====>>>>In Fig. 9, the positions of the (A) plots related to Wa and the (D) plots related to ACP were incorrect, so the figures were replaced and corrected.

====>>>>On page 4, Figure caption of Figure 3-2 was added the words “(5500-5000 cm-1)”.

Round 2

Reviewer 3 Report

Authors have addressed all my comments, I am satisfied with the revised manuscript.